# Endoscopic Diagnosis of Small Bowel Tumor

**DOI:** 10.3390/cancers16091704

**Published:** 2024-04-27

**Authors:** Tomonori Yano, Hironori Yamamoto

**Affiliations:** Department of Medicine, Division of Gastroenterology, Jichi Medical University, Shimotsuke 329-0498, Japan; tomonori@jichi.ac.jp

**Keywords:** balloon-assisted enteroscopy, capsule endoscopy, CT enterography

## Abstract

**Simple Summary:**

Recent technological advances, including capsule endoscopy (CE) and balloon-assisted endoscopy (BAE), have revealed that small intestinal disease is more common than previously thought. Early diagnosis of small intestinal tumors is essential for favorable outcomes. For early diagnosis, after examination of the upper and lower gastrointestinal tract, the possibility of small bowel lesions should be considered in patients with unexplained symptoms and signs, including gastrointestinal bleeding, chronic anemia, abdominal pain, obstructive symptoms, body weight loss, palpable abdominal mass, and fever of unknown origin.

**Abstract:**

Recent technological advances, including capsule endoscopy (CE) and balloon-assisted endoscopy (BAE), have revealed that small intestinal disease is more common than previously thought. CE has advantages, including a high diagnostic yield, discomfort-free, outpatient basis, and physiological images. BAE enabled endoscopic diagnosis and treatment in the deep small bowel. Computed tomography (CT) enterography with negative oral contrast can evaluate masses, wall thickening, and narrowing of the small intestine. In addition, enhanced CT can detect abnormalities outside the gastrointestinal tract that endoscopy cannot evaluate. Each modality has its advantages and disadvantages, and a good combination of multiple modalities leads to an accurate diagnosis. As a first-line modality, three-phase enhanced CT is preferred. If CT shows a mass, stenosis, or wall thickening, a BAE should be selected. If there are no abnormal findings on CT and no obstructive symptoms, CE should be selected. If there are significant findings in the CE, determine the indication for BAE and its insertion route based on these findings. Early diagnosis of small intestinal tumors is essential for favorable outcomes. For early diagnosis, the possibility of small bowel lesions should be considered in patients with unexplained symptoms and signs after examination of the upper and lower gastrointestinal tract.

## 1. Introduction

Small bowel tumors (SBTs) are relatively rare in incidence. They account for only approximately 3–6% of all gastrointestinal neoplasms. According to population-based cancer incidence data in the United States, the incidence of SBTs has increased over the past 20 years, from 5260 per year in 2004 to 12,440 in 2024 in the USA, and deaths due to SBTs have increased from 1130 in 2004 to 2090 in 2024 [1,2]. 

SBTs comprise different histological subtypes, including adenocarcinoma (30–45%), neuroendocrine tumors (20–40%), lymphomas (10–20%), and sarcomas (10–15%) [3]. Distribution varies geographically; in the United States, neuroendocrine tumors are most common (35–42%), followed by adenocarcinoma (30–40%) [4]. In Japan, lymphomas (47%) are the most common, followed by gastrointestinal stromal tumors (25%), and adenocarcinoma (24%) [5]. 

The term SBT often includes not only malignant neoplasms but also benign neoplasms and non-neoplastic lesions in the small bowel. In this review, the term SBT includes them.

## 2. Symptoms and Signs of Small Bowel Tumors

Recent technological advances, including capsule endoscopy (CE) and balloon-assisted endoscopy (BAE), have revealed that small intestinal disease is more common than previously thought. Although small bowel tumors are less common than in other gastrointestinal tracts, any disease is difficult to diagnose without “suspecting” it. When the following symptoms and signs are unexplained after examination of the upper and lower gastrointestinal tracts, small bowel disease may be the cause. 

Gastrointestinal bleedingAnemiaAbdominal painObstructive symptomBody weight lossPalpable abdominal massFever of unknown origin

## 3. Family History

Family history is another important clue to the diagnosis of small bowel tumors because the following hereditary diseases have an increasing risk of small bowel tumors.

Lynch syndrome (HNPCC; hereditary non-polyposis colorectal cancer) is defined by germline mutations in one of the mismatch repair (MMR) genes, mostly *MLH1*, *MSH2,* and *MSH6*. Patients with Lynch syndrome have a 100-fold increased risk of small bowel cancer compared to the general population [6].

Patients affected by neurofibromatosis type 1 (NF-1), also known as von Recklinghausen disease, have an increased risk of developing gastrointestinal stromal tumors (GIST) [7].

Familial adenomatous polyposis of the colon (FAP) is a disease of autosomal dominant inheritance and is caused by a pathogenic germline variant in the adenomatous polyposis coli (APC) gene. In patients with FAP, the cumulative risk of duodenal cancer is estimated at 4% at 70 years of age [8].

Peutz–Jeghers syndrome (PJS) is caused by germline mutations in the serine-threonine kinase 11 (*STK11*) gene (formerly known as *LKB1*) located on chromosome 19p13.3 [9]. The lifetime risk of developing cancers in a PJS patient ranges from 55% to 83% by age 60–70, including colon cancer (39%), pancreatic cancer (11–36%), and small bowel cancer (29%) [10].

## 4. Characteristics of Each Modality for Small Intestinal Tumors

### 4.1. Capsule Endoscopy (CE)

CE has advantages, including a high diagnostic yield, being discomfort-free, its outpatient basis, and physiological images. 

However, CE has several disadvantages. Because the lumen is not inflated by gas insufflation, diverticula and submucosal tumors (SMT) are often missed. Lack of irrigation and aspiration capabilities makes it difficult to detect lesions if there are a lot of residues. In patients with intestinal stenosis, there is a risk of retention. When the patency of the gastrointestinal tract cannot be confirmed, CE should not be used for patients with definitive obstructive symptoms.

CE depends on peristalsis to move, so it takes at least several hours to complete the test. CE cannot evaluate the bypassed intestinal tracts or the afferent limb after Roux-en-Y reconstruction in patients with surgically altered anatomy. 

Because CE passage is too rapid in the duodenum and the proximal jejunum, CE can recognize only 42.7% to 43.6% of the duodenal papillae [11,12]. The lesions in the duodenum and the proximal jejunum can be missed by CE. Han et al. reported that small bowel tumors were not detected by CE but were eventually diagnosed by DBE in nine (16.7%) of 54 patients. Five lesions (55.6%) of the nine missed lesions were located in the proximal jejunum [13].

When there are multiple similar lesions, it is difficult to distinguish each and count the lesions. CE is not suitable for counting lesions in polyposis syndromes.

Even large tumors can be missed by CE. If the CE is caught on the proximal side of a large tumor, the tumor may not be detected, depending on the direction of the CE’s camera. After staying for a while, the CE quickly slips through the large tumor area, and the large tumor is not captured by the CE. If the CE remains in the same location for more than 15 min, this is an abnormal finding known as a regional transit abnormality (RTA) [14].

CE can be very useful if it is used with an understanding of the above characteristics.

### 4.2. Balloon-Assisted Endoscopy (BAE)

The double-balloon endoscopy (DBE) is equipped with two balloons attached at the endoscope’s tip and the overtube’s tip. It enabled endoscopic diagnosis (Figure 1) and treatment in the deep small bowel. The single-balloon endoscopy (SBE), developed after DBE’s launch, omits the balloon at the tip of the endoscope to simplify the preparation. Both are collectively referred to as balloon-assisted endoscopies (BAE). The BAE can be inserted with the aid of a ballooned overtube to suppress unnecessary deflection of the intestine. In addition, because the bowel is folded over the overtube, the BAE can be inserted into intestinal tracts longer than the working length of the scope. BAE can be inserted without relying on intestinal peristalsis, allowing evaluation of the afferent limb and bypassed intestinal tracts.

The BAE is equipped with a working channel that allows for procedures such as tissue biopsy, marking, and treatment. Some kinds of small bower tumors can be treated by endoscopic treatment or chemotherapy. Before starting the chemotherapy, its histological diagnosis should be confirmed. When a small bowel tumor requires surgical treatment, endoscopic tattooing facilitates laparoscopic surgery. Chromoendoscopy with indigo-carmine makes it easy to detect small lesions in patients with FAP [15]. The miniature probe can be inserted into the working channel and enables endoscopic ultrasound evaluation for submucosal tumors [16]. 

In the setting of X-ray fluoroscopy, endoscopic enteroclysis can be performed by injecting contrast through the working channel. The size and shape of the lesion can be evaluated by fluoroscopy. During endoscopic enteroclysis, the scope balloon of DBE can inflate to reduce the backflow of contrast and evaluate the wide range of the small bowel by fluoroscopy.

BAE has several disadvantages. BAE requires endoscopic skills. Severe adhesions or stenosis make it difficult to achieve total enteroscopy with BAE. Especially near large tumors, maneuverability may be poor due to compression and adhesions caused by the tumor, and it may not be possible to reach the lesion.

### 4.3. Computed Tomography (CT)

Recent technological advances have increased the usefulness of computed tomography (CT) in the diagnosis of small bowel lesions. Since the introduction of MDCT (Multi-Detector-Row CT) with 4-row detectors in 1998, the number of detectors has increased and evolved to 16-row, 64-row, and 320-row. As a result, images with high spatial resolution can be obtained in a short time over the entire abdomen. In addition to axial section, coronal and sagittal section images can be reconstructed, and multiplanar reformation (MPR) images, virtual enteroscopy, and virtual enteroclysis are also available.

CT can evaluate ascites, misty mesentery, and abnormal blood vessels. In diagnosing small bowel tumors, CT is very useful in evaluating lymphadenopathy associated with malignant lymphoma and small intestinal cancer, as well as extra-luminal GIST, which is difficult to evaluate by endoscopy. However, plain CT provides a very limited amount of information and makes it difficult to detect lesions, so it is preferable to use at least contrast-enhanced CT and, if possible, dynamic CT. Some kinds of small bowel tumors are difficult to detect with conventional contrast-enhanced CT and are easily detected with dynamic CT. Shinya et al. reported that gastrointestinal tumors and neuroendocrine tumors demonstrated a hyper-vascular pattern in the multiphasic dynamic CT. Adenocarcinomas and lymphomas showed a delayed enhancement pattern [17].

One problem with CT is that CT images are momentary images, and depending on the timing of imaging, the shape of the intestinal tract due to peristalsis or spasm may appear as a stenosis or mass. Dynamic CT can solve this problem by comparing the intestinal geometry between images taken at different times (plain, early contrast, and late contrast). Dynamic CT makes it easier to distinguish intestinal stricture from peristalsis and spasms of the intestinal tract. 

CT enterography, in which a bowel cleansing medium such as polyethylene glycol is taken as a negative contrast agent before the CT scan, provides detailed imaging of the small bowel by adequate lumen distention and provides information on masses, wall thickening, and narrowing of the small bowel. According to a meta-analysis, the sensitivity and specificity of CT enterography for small bowel tumors were 0.93 and 0.83, respectively [18]. Although there are problems with radiation exposure and side effects from contrast media, it is a minimally invasive test that provides a large amount of information quickly. Magnetic resonance enterography has similar sensitivity and specificity [18] and can be an alternative with no radiation exposure when it is available.

## 5. Diagnostic Strategy for Small Bowel Tumors

Each modality has its advantages and disadvantages, and a good combination of multiple modalities leads to an accurate diagnosis since a false negative or false positive result is possible with a single modality alone.

Honda et al. reported a comparative study of the diagnostic yields of contrast-enhanced CT, fluoroscopic enteroclysis, CE, and DBE for small bowel tumors [19]. In their comparing study, diagnostic yields for small bowel tumors </=10 mm were significantly low in contrast-enhanced CT and fluoroscopic enteroclysis. However, the diagnostic yields of CE and DBE were high for small bowel tumors, regardless of size. In contrast-enhanced CT, the diagnostic yield of epithelial tumors was significantly lower compared with subepithelial tumors. The diagnostic yields of CE and DBE were significantly higher than those of contrast-enhanced CT, and the diagnostic yield of DBE was significantly higher than that of CE. However, a combination of contrast-enhanced CT and CE had a diagnostic yield similar to that of DBE. Because CE and CT can cover each other’s shortcomings in detecting small bowel tumors, the combination use of CE and contrast-enhanced CT is recommended for detecting small bowel tumors. After the screening, DBE is useful for histologic diagnosis and endoscopic treatment.

Based on the results of the above study, we recommend the following strategies (graphic abstract).

As a first-line modality, dynamic CT is preferred because CT findings are informative to select the next test, CE or BAE. In addition to axial images, coronal images should be produced for precise reading. CT enterography makes it easier to detect masses in the small intestine. In clinical practice, CT and colonoscopy can be scheduled on the same day, and CT can be taken before the colonoscopy to obtain CT enterography images. 

If CT shows a mass, stenosis, or wall thickening, a BAE should be selected because of its capability for biopsy and marking. The route of insertion of the BAE should be selected based on the information obtained from the CT.

If there are no abnormal findings on CT and no obstructive symptoms, CE should be selected. If there are significant findings in the CE, determine the indication for BAE and its insertion route based on these findings. However, CE often misses diverticula and submucosal tumors due to its inability to insufflate gas. It can also miss lesions in the duodenum and the proximal jejunum due to its rapid movement. CE cannot evaluate the bypassed intestinal tracts or the afferent limb in patients with surgically altered anatomy. After negative CE, the indication of further examinations should be decided with an understanding of the above characteristics of CE.

## 6. The Role of Enteroscopy in Each Disease

### 6.1. Small Bowel Adenocarcinoma

The rate of primary small bowel adenocarcinoma is less than 3% of all gastrointestinal cancers. Risk factors for small bowel adenocarcinoma include FAP, HNPCC, PJS, Crohn’s disease, and celiac disease.

Most primary small bowel adenocarcinomas arise in the duodenum, the proximal jejunum, or the distal ileum. They are often found with obstructive symptoms or chronic iron deficiency anemia. 

Although they can sometimes be reached with a conventional endoscope, the range of routine upper gastrointestinal endoscopy and colonoscopy does not include the deep duodenum, the proximal jejunum, or the distal ileum. To endoscopically diagnose small bowel adenocarcinoma, intentional deep insertion is necessary. As a result, at the time of diagnosis, most of the patients were in an advanced stage with metastasis to other organs or peritoneal dissemination. The multi-center retrospective study, which included 354 patients with primary small bowel adenocarcinoma, reported that the rates for clinical stages 0, I, II, III, and IV at the time of diagnosis were 5.4%, 2.5%, 27.1%, 26.0%, and 35.6%, respectively [5]. The tumor stage is the most important prognostic factor for small bowel adenocarcinoma. Therefore, for early diagnosis, the possibility of small bowel lesions should be considered in patients with unexplained symptoms and signs after routine upper gastrointestinal endoscopy and colonoscopy.

BAE, or push enteroscopy, can reach the lesion (Figure 1), take a biopsy for histopathologic diagnosis, and mark it by tattooing for surgical treatment. Endoscopic findings of small bowel adenocarcinoma often include ulceration and stenosis. Type 2 (54.2%) was the most common among the macroscopic types, followed by Type 3 (18.2%) [5].

### 6.2. Gastrointestinal Stromal Tumor

Gastrointestinal stromal tumors (GIST) are often caused by gastrointestinal bleeding but can also be found incidentally on contrast-enhanced CT for the evaluation of other diseases. GIST is a mesenchymal malignancy derived from the interstitial cells of Cajal that control intestinal peristalsis. Patients with NF-1 are often associated with GISTs. NF1-associated GISTs occur in younger patients compared with sporadic GISTs and often multiple tumors, mainly incidental, localized at the small bowel and in the absence of KIT/PDGRF*α* mutations [20,21].

GIST growth patterns include extraluminal, intraluminal, or mixed (dumbbell-shaped) patterns. GIST is a submucosal tumor covered by normal mucosa. Because it is difficult to distinguish from extraluminal compression, GISTs are often missed by CE. Ulcers/erosions or dilated abnormal vessels are important findings for detecting GIST using CE.

Although GIST with intraluminal and mixed patterns can be detected by BAE (Figure 2), GIST with extraluminal patterns is hardly detected by endoscopy, except for abnormal vessels and unnatural traction findings due to lesions. 

Tattooing by BAE is helpful for identifying the lesion during laparoscopic-assisted partial resection of the small intestine. 

Endoscopic ultrasound fine needle aspiration (EUS-FNA) became a routine examination for evaluating gastric GIST. However, EUS-FNA is not available for small-bowel GIST due to a lack of dedicated equipment. Endoscopic biopsy for GIST has a low diagnostic rate and cannot determine malignancy accurately due to its subepithelial nature [22,23]. Endoscopic biopsy also carries the risk of post-biopsy bleeding. Biopsy is indicated only when contrast-enhanced CT or endoscopic findings are atypical or when histopathology is necessary prior to chemotherapy for unresectable lesions.

Symptomatic GISTs, regardless of size, are indicated for surgical resection if they are resectable. The indication for surgical treatment for asymptomatic GISTs is determined by their size and rate of growth. Small-bowel GISTs have a significantly higher metastatic risk than gastric GISTs [20]. However, for multiple GISTs in NF-1 patients, small lesions may be followed up without surgical resection since they have favorable histologic parameters (relatively low mitotic rates) [24], and it is difficult to resect all multiple lesions.

### 6.3. Malignant Lymphoma

Malignant lymphomas in the small intestine have been conventionally diagnosed by radiographic examinations, and surgical resections were required for histological diagnosis. BAE enables tissue biopsy and histopathological diagnosis of primary small intestinal lymphoma without surgical resection. 

The macroscopic findings of small bowel lymphoma are classified as polypoid, ulcerative (including stricturing, non-stricturing, and aneurysmal forms), polyposis (multiple lymphomatous polyposis), diffusely infiltrating, or mixed type. There is some correlation between the macroscopic and histological types. Many cases of ulcerative type are histologically diffuse large B-cell lymphoma (DLBCL). Most cases of polyposis type (multiple lymphomatous polyposis) are follicular lymphoma or mantle cell lymphoma, while diffusely infiltrating type tends to comprise either T-cell lymphomas or immunoproliferative small intestinal disease [25].

Although endoscopic findings of malignant lymphomas vary by histologic type (Figure 3), a definitive histopathologic diagnosis can be made by biopsy in most cases. Based on the histopathologic diagnosis, lymphomas can be treated with chemotherapy [26,27]. However, in cases of bleeding or obstructive symptoms, chemotherapy is given after surgical treatment. 

Endoscopic tattooing is useful for recognizing lesions during surgical treatment and for identifying lesion sites for follow-up BAE after chemotherapy when complete remission is achieved.

Endoscopic balloon dilation for post-chemotherapy stenosis is an alternative option that avoids surgical treatment [28]. 

### 6.4. Neuroendocrine Tumor

Gastrointestinal neuroendocrine tumors (GI NET), formerly known as carcinoids. According to the WHO classification, NETs are classified into low-grade neuroendocrine tumors (NETs) and high-grade neuroendocrine carcinomas (NECs). NETs are also classified into G1, less malignant, and G2, more malignant than G1. 

The secretion of serotonin and other hormones from NET can cause facial flushing, diarrhea, bronchospasm, and other symptoms known as carcinoid syndrome. The carcinoid syndrome is more frequent in NET, especially those occurring in the jejunum, ileum, and appendix, than in other gastrointestinal sites.

Multifocal NET may occur in 30–50% of patients. Patients with multiple lesions are younger than those with solitary tumors, have a significantly higher risk of developing carcinoid syndrome, and have a poorer prognosis [29].

The endoscopic image of NET is yellowish SMT-like, but it is precisely a tumor of epithelial origin. It is often seen as an elastic, hard, mobile tumor with atrophied surface villi and dilated capillaries (Figure 4a). Depressions, ulcers, or erosions on the surface may indicate a high-grade lesion (Figure 4b).

Although CE may be useful in identifying NETs, it cannot confirm the correct number of multiple lesions or perform marking at the lesions.

The sensitivity of BAE for the primary SB-NET was 88%, compared to 60% for CT, 54% for MRI, and 56% for somatostatin receptor imaging. BAE could also be considered for detecting multifocal NETs before surgery. In patients who underwent small bowel resection, additional lesions were found in 54% of patients at preoperative BAE, but only 18% of patients at preoperative CE [30].

Endoscopic resection is not recommended for jejunal and ileal NETs due to the risk of invasion and lymphatic spread, even with diminutive lesions, which may necessitate a more extensive surgical resection.

### 6.5. Metastatic Tumors

Metastatic tumors of the small bowel include direct invasion from other organs, intraperitoneal disseminated tumors, and metastatic tumors due to hematogenous metastasis. Pancreatic cancer frequently directly invades the duodenum. Colon, ovary, uterus, and stomach cancers metastasize to the small intestine through direct invasion or intraperitoneal dissemination. Lung cancer, breast cancer, and malignant melanoma metastasize hematogenously to the small intestine. The most common primary tumor for metastatic tumors in the small intestine is lung cancer.

The endoscopic image of the metastatic lesion is variable (Figure 5). Metastatic lesions can present as single or multiple polypoid lesions, with or without ulcers. Metastatic tumors arise from the submucosa and are sometimes difficult to distinguish from malignant lymphomas on endoscopic images. They may be seen as focal bowel wall thickening and cause luminal narrowing. 

The management of the metastatic lesion depends on the symptoms and stage of the primary tumor. Surgical resection of the affected intestine can be useful to relieve symptoms such as obstruction and bleeding.

### 6.6. Benign Tumors

Many benign tumors of the small intestine are asymptomatic, making it difficult to calculate their exact prevalence. Benign tumors can cause overt or obscure bleeding with chronic anemia. Larger tumors can cause obstructive symptoms due to a narrowing of the lumen or intussusception. In the past, they were often found by chance during surgery or autopsy of other diseases, but with the widespread use of CE and BAE, there are more and more opportunities for diagnosis by endoscopy.

Benign small bowel tumors include adenomas, hamartomas, lipomas (Figure 6a), hemangiomas (Figure 6b), lymphangiomas (Figure 7a,b), and inflammatory fibroid polyps (Figure 8a). The ectopic pancreas can be found as a submucosal tumor-like lesion (Figure 8b). 

Adenoma can be treated by various techniques, such as cold snare polypectomy (CSP), endoscopic mucosal resection (EMR), underwater endoscopic mucosal resection (UEMR) [31,32], gel immersion endoscopic mucosal resection (GIEMR) [33,34,35,36,37,38], and endoscopic submucosal dissection (ESD). Hamartomas [39], lipomas [40,41], lymphangiomas [42], and inflammatory fibroid polyps [38] can also be resected endoscopically. Hemangioma, especially in patients with blue rubber bleb nevus syndrome [43], can be treated by various techniques, such as polidocanol injection therapy [44], electro-coagulation [45], polypectomy [46], band-ligation [47], and loop-ligation [48]. Of course, surgical treatment should be considered for massive lesions, even benign tumors.

### 6.7. Peutz–Jeghers Syndrome

Peutz–Jeghers syndrome (PJS) is an inherited polyposis syndrome characterized by the presence of multiple hamartomatous polyps (Figure 9) throughout the gastrointestinal tract, excluding the esophagus. It is accompanied by mucocutaneous melanin pigmentation and an elevated lifetime risk of both gastrointestinal and extra-gastrointestinal malignancies [10,49]. PJS is inherited in an autosomal dominant manner, yet around 45% of PJS patients are de novo cases. The estimated incidence has been reported at 1/200,000 [50].

The malignant potential of polyps is low, but the polyps can enlarge, resulting in intussusception and emergency laparotomy. Polyps can develop and grow throughout life, and repeated surgical treatment can lead to short bowel syndrome. Because intra-abdominal adhesions due to surgery can cause difficulty in total enteroscopy with BAE, the digestive tract should be examined, and endoscopic treatment with BAE should be initiated before emergency laparotomy for intussusception. In patients with PJS, gastrointestinal surveillance through upper gastrointestinal endoscopy, colonoscopy, and CE should begin by the age of 8 years old at the latest. European and Japanese guidelines recommend that SB polyps > 15 mm be treated to prevent intussusception [51].

The conventional techniques for polyp removal in PJS are snare polypectomy and endoscopic mucosal resection [52,53]. However, these conventional techniques have a risk of complications such as perforation and bleeding. Recently, endoscopic ischemic polypectomy (EIP) has been described, which involves strangulating the stalk of a polyp using a detachable snare [54] or clips [55] to induce polyp destruction. Performing EIP with clips is technically easier compared to conventional techniques because EIP requires visualization of only the stalk of the polyp, even within a limited working space. The advantages of EIP include the removal of a larger number of polyps and the prevention of complications after polypectomy, such as bleeding, perforation, or post-polypectomy syndrome [56,57]. Considering the low risk of adverse events in EIP, EIP has the potential to change the size threshold for endoscopic treatment. The disadvantage of EIP is a lack of histopathological evaluation for treated polyps. 

Endoscopic reduction of intussusception in patients with PJS is a viable alternative to surgery, except for patients with necrosis or perforation. The reported success rate of endoscopic reduction of 22 sites in 19 patients was 95%, with only two mild pancreatitis adverse events [58].

### 6.8. Familial Adenomatous Polyposis

Familial adenomatous polyposis (FAP) is a rare genetic predisposition primarily to digestive cancers, inherited in a dominant manner (APC gene) or recessive manner (MUTYH gene) for the main types of FAP. The definition of classical FAP is based on the presence of at least 100 colorectal adenomas [59]. Adenomatous polyps in the duodenum are found in nearly 100% of individuals with classical APC-related FAP and in approximately 30% of patients with biallelic MUTYH mutations, and less frequently in the distal small intestine [60]. In patients with FAP, other than colorectal cancer, duodenal cancer and desmoid tumors are significant contributors to mortality [61]. 

A systematic gradient from the duodenum to the jejunum is observed, as indicated by the low risk of small bowel cancer as far as the distal small bowel is considered [62]. Endoscopic surveillance for duodenal adenomas is recommended to start around 25 years old [6,63]. Chromoendoscopy with indigo carmine is useful in detecting small lesions (Figure 10a). A large adenoma in the proximal jejunum is sometimes detected by double-balloon enteroscopy after a long period of surveillance and treatment using only conventional upper gastrointestinal endoscopy (Figure 10b) [64]. Endoscopic surveillance for jejunal adenomas should be considered at several-year intervals, depending on the severity of the patient. Chromoendoscopy with indigo carmine can increase the detection of adenomas [15,65].

In the recent ESGE guideline, endoscopic treatment of lesions in the duodenum and jejunum has been recommended for lesions larger than 10 mm in size [6] because of endoscopic maneuverability and the risk of complications. However, the size threshold for endoscopic resection should be optimized by weighing the risks and benefits. The feasibility and safety of performing a cold snare polypectomy for duodenal adenomas in patients with FAP were reported [66]. CSP enabled the removal of a greater number of polyps in a shorter duration while maintaining safety [67]. Underwater endoscopic mucosal resection [31] for sporadic nonampullary duodenal adenoma [32] is reported as a safe and effective procedure. These new techniques have the potential to change the size threshold for endoscopic resection. Takeuchi et al. reported that intensive downstaging polypectomy using the new techniques showed significant downstaging with acceptable adverse events for multiple duodenal adenomas in patients with FAP [68].

Each adenoma may be fused in patients with multiple adenomas and become a larger lesion. Endoscopic resection of larger adenomas may increase the risk of adverse events. At least in patients with a large number of adenomas, intensive downstaging polypectomy should be considered, even without adenomas larger than 10 mm. 

## 7. Conclusions

Advances in various medical technologies have greatly advanced the diagnosis and treatment of small intestinal diseases. However, small intestinal cancer is often found at an advanced stage. Early diagnosis of small intestinal tumors is essential for favorable outcomes. For early diagnosis, the possibility of small bowel lesions should be considered in patients with unexplained symptoms and signs after examination of the upper and lower gastrointestinal tract.

## Figures and Tables

**Figure 1 cancers-16-01704-f001:**
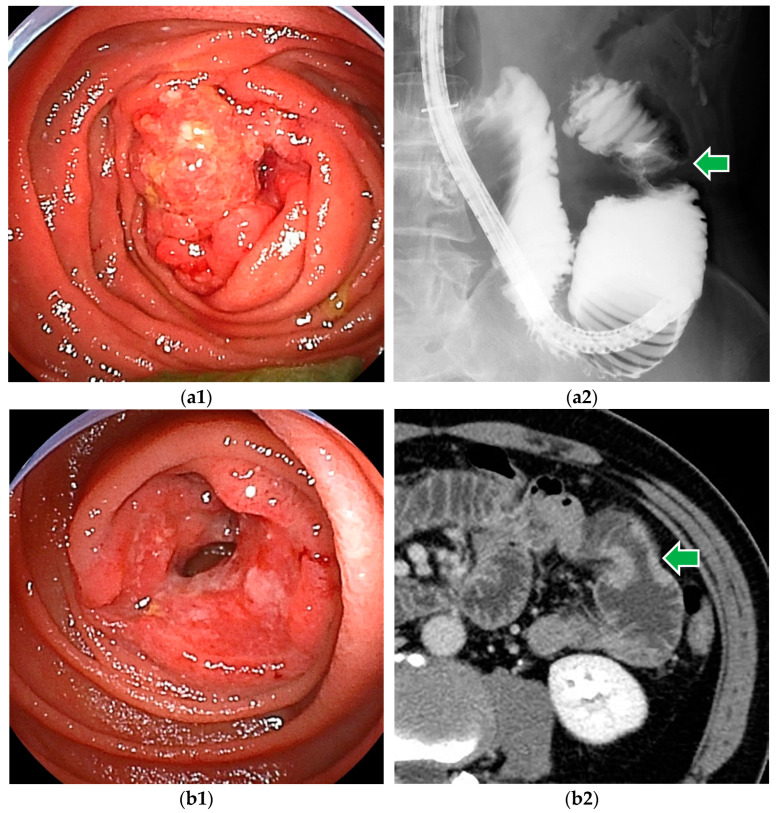
Small bowel adenocarcinomas, which are often advanced at the time of diagnosis, and endoscopic findings often include ulceration and stenosis. (**a1**,**a2**) DBE revealed adenocarcinoma in the proximal jejunum. Endoscopic enteroclysis showed the stenosis as an apple core sign. (**b1**,**b2**) DBE revealed adenocarcinoma with ulceration. CT showed mild stenosis.

**Figure 2 cancers-16-01704-f002:**
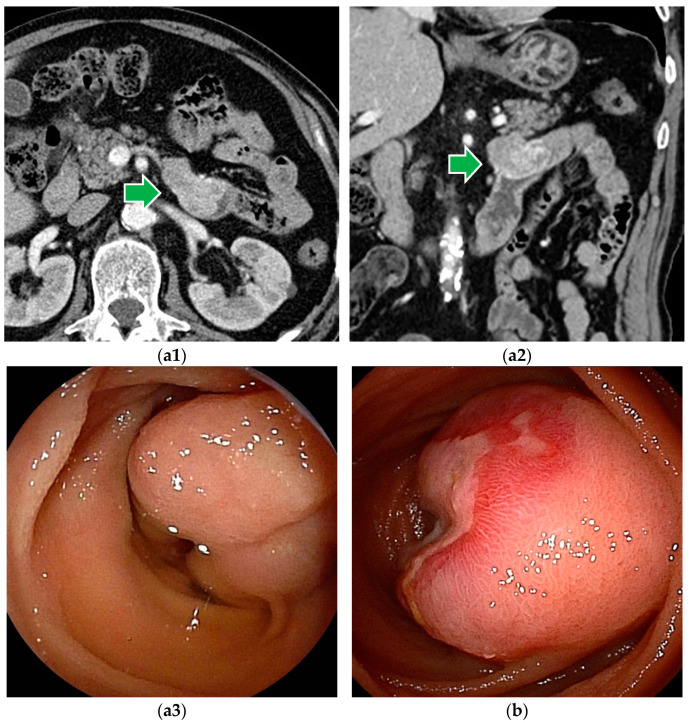
Gastrointestinal stromal tumor (GIST): (**a1**–**a3**) Contrast-enhanced CT revealed a well-enhanced lesion. DBE showed a submucosal tumor covered by normal mucosa. (**b**) In patients with bleeding symptoms, erosions, ulcers, or dilated blood vessels are seen on the surface.

**Figure 3 cancers-16-01704-f003:**
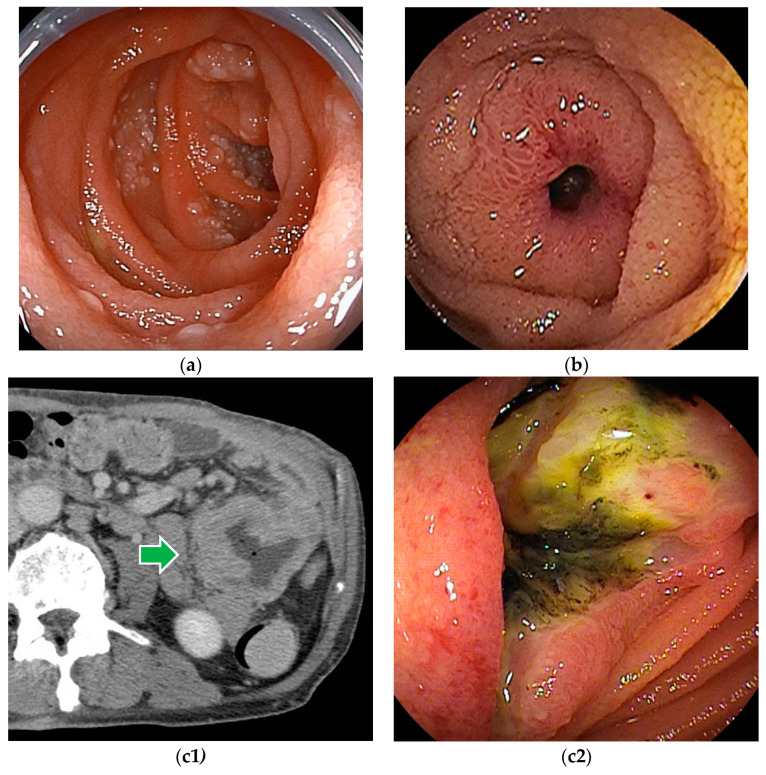
Malignant lymphoma: (**a**) Follicular lymphoma is characterized by aggregations of large and small white granules. The lesions are distributed focally from the duodenum to the jejunum. (**b**) Rarely, it may be a form of concentric stenosis with ulceration. (**c1**,**c2**) Diffuse large B-cell lymphoma (DLBCL) often shows an ulcerated or polypoid morphology, and the biopsy should be taken from the ulcer bed rather than from the edges. CT revealed a wall-thickened intestine with a dilated lumen. DBE showed an ulcerated lesion.

**Figure 4 cancers-16-01704-f004:**
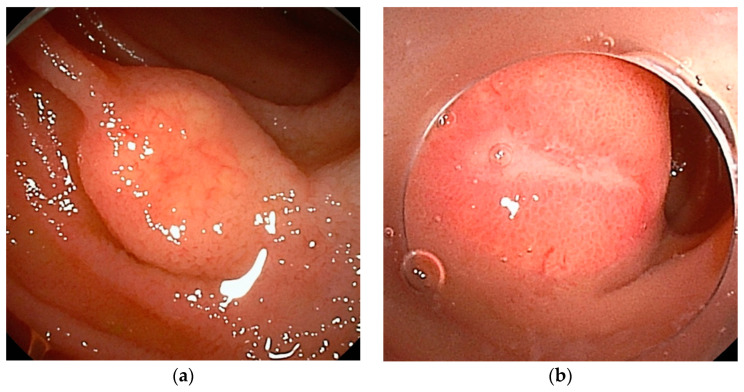
Neuroendocrine tumor: (**a**) SMT-like lesion with atrophied surface villi and dilated capillaries (**b**) A high-grade lesion with ulcers on the surface.

**Figure 5 cancers-16-01704-f005:**
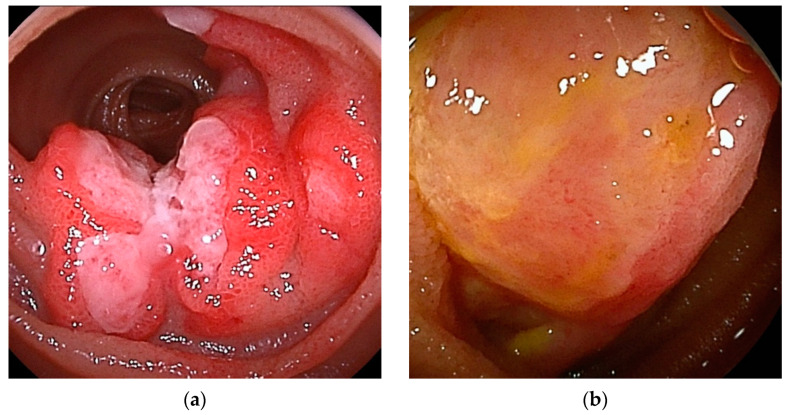
Metastatic tumors: (**a**) metastatic jejunal tumor from angiosarcoma of the breast. (**b**) Metastatic jejunal tumor from lung cancer.

**Figure 6 cancers-16-01704-f006:**
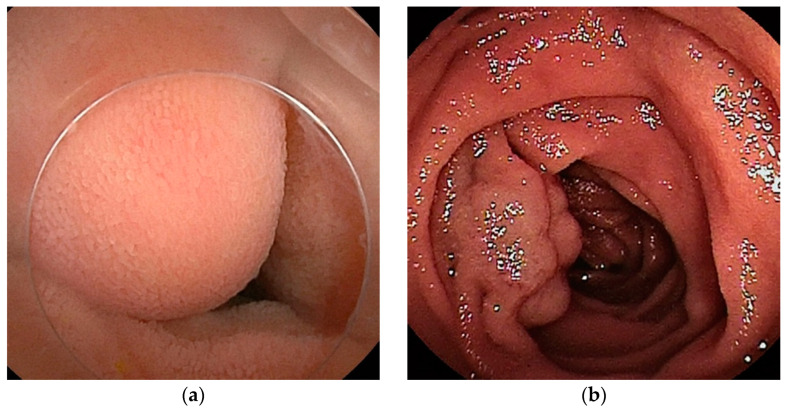
(**a**) Lipoma: Lipomas are yellowish-white submucosal tumors that are soft and deform when pressed with forceps. It is characterized by low density on CT as well as fatty tissue and high echoic lesions on ultrasound. (**b**) Cavernous hemangioma: Localized cavernous hemangiomas are soft, pale to dark red submucosal tumors with a smooth surface.

**Figure 7 cancers-16-01704-f007:**
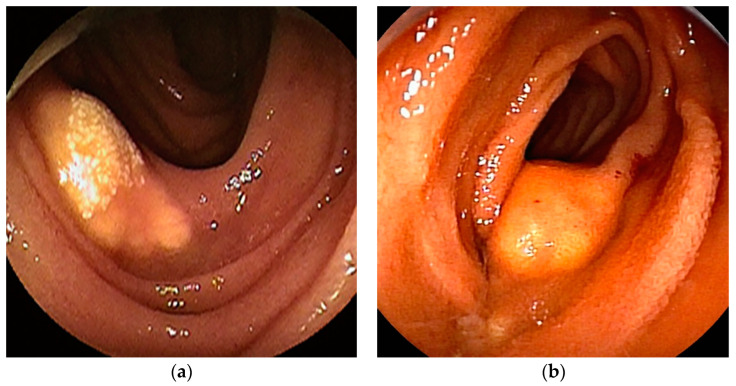
Lymphangioma: Lymphangiomas have different endoscopic appearances depending on the depth of the dilated lymphatic channels. (**a**) If the lesion has dilated lymphatic channels in the mucosa, it will be an elevation with white dots on the surface. (**b**) If the lesion is primarily in the submucosa, it will have a smooth surface and a yellowish-white to pale blue submucosal tumor without white dots.

**Figure 8 cancers-16-01704-f008:**
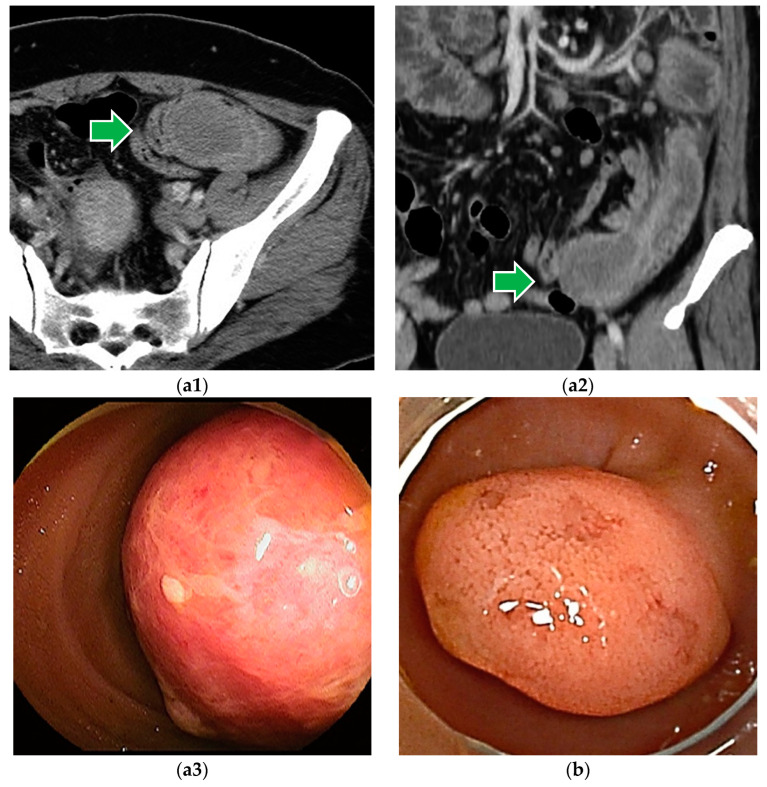
(**a1**–**a3**) Inflammatory fibroid polyp (IFP): IFP is a pedunculated or sub-pedunculated lesion that may be found as a submucosal tumor, but as it grows, the mucosa is shed by mechanical stimulation and is found as a smooth, protruding lesion. A large IFP can cause intussusception. This lesion was treated by a laparoscopy-assisted partial small bowel resection. (**b**) Ectopic pancreas: Ectopic pancreas presents as a 10~20 mm-sized, hemispherical, or sub-pedunculated SMT-like appearance with multiple nodules covered by thin, normal mucosa reflecting the internal multifocal structure, with a slight depression on the surface.

**Figure 9 cancers-16-01704-f009:**
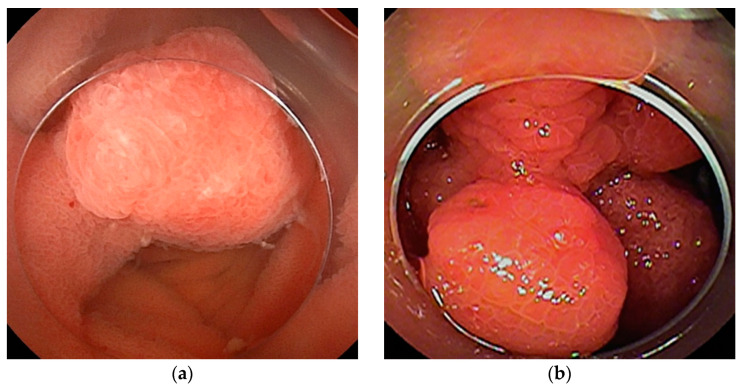
Polyps in patients with Peutz–Jeghers syndrome: (**a**) Most polyps in patients with Peutz–Jeghers syndrome are pedunculated or sub-pedunculated polyps. (**b**) Some of the growing lesions are branched, bifid, or multinodular, reflecting dendritic growth of the muscularis mucosa.

**Figure 10 cancers-16-01704-f010:**
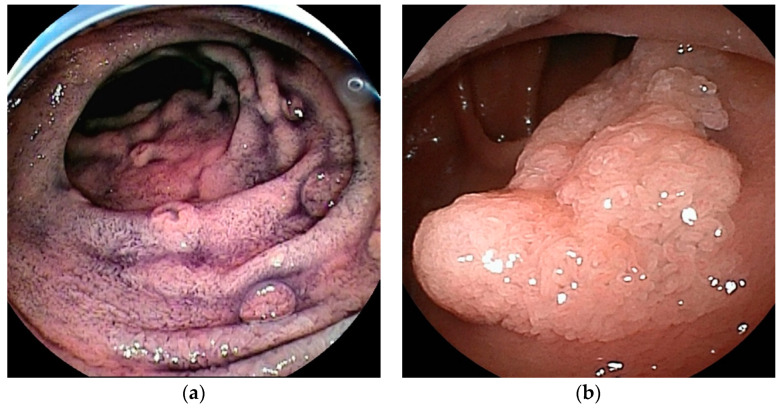
Polyps in patients with familial adenomatous polyposis: (**a**) Small, flat lesions often occur from the duodenum to the jejunum. (**b**) A large adenoma in the proximal jejunum was detected by double-balloon enteroscopy after a long period of surveillance and treatment using only conventional upper gastrointestinal endoscopy. This lesion was treated by endoscopic submucosal dissection (ESD) using the pocket-creation method and balloon-assisted endoscopy [64].

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
