# Peer review of "Endoscopic Diagnosis of Small Bowel Tumor"

_cancers, 2024, doi:10.3390/cancers16091704_

Round 1
Reviewer 1 Report
Comments and Suggestions for Authors
an interesting subject
well organized material
histology may be also important but the fact that the tumor is usually either obstructive or bleeding makes the surgical intervention also necessary
Reviewer 2 Report
Comments and Suggestions for Authors
This article is very interesting, but it contains some problems (see below). It is not acceptable for publication in the present form.
Major points
1) The explanation is too short and superficial throughout the manuscript.
5.3. Malignant lymphoma: Please describe more about follicular lymphoma, diffuse large B-cell lymphoma, and please describe the Figures in more detail.
5.6. Benign tumors: only 11 lines without Figure. Please describe each disease in more detail.
2) The authors compare very roughly diagnostic performance of CT, CE, and BAE. Please analyze more strictly many previous papers.
Minor points
3) Abbreviation: Please add a list of abbreviations. It will help read smoothly the manuscript.
4) Diagnostic algorithm: Please add a description of the diagram.
5) Figures: Only one Figure in this review. Please add more Figures. It will help the readers understand this Review more deeply.
Reviewer 3 Report
Comments and Suggestions for Authors
The aim of this study is to review the world literature relative to the endoscopic diagnosis of small bowel tumor.
Maior issues:
The title „Endoscopic diagnosis of small bowel tumor” does not precisely correlate with the content of this manuscript, because other diagnostic methods such as computed tomography are also described in this paper. There are review papers regarding endoscopic diagnostics of small bowel tumors in the world literature. This work is not an innovative and crucial contribution to the field. There are not wide spectrum and critical literature review (only 37 references) on the role of endoscopic investigations in the diagnostic process in patients with small bowel tumor in this manuscript.
Minor issues:
1. Introduction section, including information on definition, types, and epidemiology of small bowel tumors, should be added.
2. Numerous terms used in the mansuccript should be corrected and precised, i.a. in the title of section „Symptoms and signs”, „small bowel tumors” should be added, abbreviation „GI” should be explained in the place of first use in the main text.
Comments on the Quality of English Language
Minor editing of English language required.
Reviewer 4 Report
Comments and Suggestions for Authors
It is a great honor to be in charge of the review of an article entitled “Endoscopic diagnosis of small bowel tumor” by Professors Yano and Yamamoto, Japanese gastroenterologists well known for their original procedure of the double-balloon endoscopy. My concerns are as follows.
1. Although the illustration entitled “Diagnostic algorithm for small bowel tumors” appearing in the beginning of this manuscript might be a key information in this review article, its explanation is noted just in six sentences in the end of 4. Diagnostic strategy for small bowel tumors. More practical and detailed statement would be appreciated. Although the term DBE is used in the algorithm, the term BAE is mainly used in Simple Summary and the main text. As it is described “Both (SBE and DBE) are collectively referred to as balloon-assisted endoscopy (BAE), it might be better to replace DBE by BAE in the algorithm. The original word of FOB, that is “fecal occult blood”, is better to be added somewhere.
2. As noted in Simple Summary, it is important “For early diagnosis, the possibility of small bowel lesions should be considered in patients with unexplained symptoms and signs after examination of the upper and lower gastro-intestinal tract”. Therefore, I consider that more details of symptoms indicative of small bowel tumors, such as obscure gastrointestinal bleeding, are better to be described.
3. Because this manuscript was submitted to the Special Issue: The Application of Endoscopy in Gastrointestinal Cancers” in Cancers, more descriptions especially regarding small bowel cancer would be appreciated.
4. Although four endoscopic photos, including small intestinal cancer, gastrointestinal stromal tumor, follicular lymphoma and B-cell lymphoma, are shown in Figure 1, I would like to ask the authors to add more of endoscopic photos with kind explanation in each pathological condition as well as small bowel cancer. It would be better if the authors could add dynamic CT images matched with each endoscopic photo obtained before BAE.
Comments on the Quality of English LanguageNil
Round 2
Reviewer 2 Report
Comments and Suggestions for Authors
This manuscript has been sufficiently revised, and it is suitable for publication in Cancers.
Author Response
Thank you.
Reviewer 4 Report
Comments and Suggestions for Authors
I truly appreciate the authors' efforts to have added valuable endoscopic pictures of small intestinal tumors in the revised version.
Although this manuscript was submitted to the Special Issue: The Application of Endoscopy in Gastrointestinal Cancers” in Cancers, it might be better if the authors could add dynamic CT images matched with each endoscopic photo obtained before BAE as much as possible, as noted in my previous review.
Comments on the Quality of English LanguageNil
Author Response
Thank you for your valuable comment. We added several CT images and endoscopic enteroclysis.